# VARIATIONAL DIFFUSION CHANNEL DECODING: A ULTRA-LOW-COST NEURAL CHANNEL DECODER

## ABSTRACT

Neural channel decoder, as a data-driven channel decoding strategy, has shown very promising improvement on error-correcting capability over the classical methods. However, the success of those deep learning-based decoder comes at the cost of drastically increased model storage and computational complexity, hindering their practical adoptions in real-world time-sensitive resource-sensitive communication and storage systems. To address this challenge, we propose an efficient variational diffusion model-based channel decoder, which effectively integrates the domain-specific belief propagation process to the modern diffusion model. By reaping the low-cost benefits of belief propagation and strong learning capability of diffusion model, our proposed neural decoder simultaneously achieves very low cost and high error-correcting performance. Experimental results show that, compared with the state-of-the-art neural channel decoders, our model provides a feasible solution for practical deployment via achieving the best decoding performance with order-of-magnitude ($1000\times$ and up) savings in computational cost and model size.

## 1 INTRODUCTION

Channel coding has served as the fundamental and critical mechanism in numerous modern communication and storage systems and applications, such as 5G, Wi-Fi, Starlink, optical networking, solid-state drive and hard disk drive. By providing error correction functionality, channel coding aims at protecting information from various corruptions (e.g., noise) incurred by data transmission. To that end, most channel codes are designed by adding extra redundant bits to help detect and recover the original information after noisy transmission.

To date, most of the commercially adopted channel codes are linear block codes, which can be optimally decoded using maximum likelihood (ML) decoding process. However, ML decoding, which can be mathematically modeled as searching for the closest lattice point in high dimensional space (Gowaikar & Hassibi, 2007), is very expensive and computationally prohibitive. In practice, a more feasible and practical channel decoding solution is to use belief propagation (BP) algorithm (Su et al., 2022), which can achieve exact optimum results in the tree-structured factor graph. However, in the context of channel coding, factor graphs constructed from the parity check matrix of modern channel codes are often cyclic, making the BP decoding results suboptimal (Yedidia et al., 2005).

**Existing Neural Channel Decoders.** Motivated by the unprecedented success of neural networks in many fields, many recent efforts leverage the advance of deep learning to develop neural channel decoder, successfully improving the decoding performance. Nachmani et al. (2016) propose a neural belief propagation model by adding neural weights on all propagation messages. Considering belief propagation is naturally a graph based algorithm, Nachmani & Wolf (2019) further improves by using hyper-graph neural network as the decoding solution. As these models are constrained to the original belief propagation form, they are not as flexible as modern neural networks in terms of layer design, architecture search, etc. To overcome this limitation, Bennatan et al. (2018) reformulates the decoding problem into a noise prediction task taking extra steps in pre-processing and post-processing, enabling more relaxed model design. Following this philosophy, Choukroun & Wolf (2022) introduces the powerful transformer (Vaswani et al., 2017) architecture-based neural channel decoder, and later extend it into a diffusion-based model (Choukroun & Wolf, 2023).

**High-Complexity Challenges of SOTA Neural Decoders.** While today's deep learning-based decoding algorithms achieve outstanding error-correcting performance, they are meanwhile suffering from the high computation and storage costs of neural networks, a very challenging limitation severely hinders their practical deployment. More specifically, the application scenarios of channel coding, *e.g.*, wireless communication, optical communication and disk drives, demand real-time and low-power processing, bringing very stringent requirement on processing speed and power consumption of channel decoder. For instance, the decoding latency in 5G is limited to millisecond level (Parvez et al., 2018; Rico & Merino, 2020). Meanwhile, a massive amount of channel decoding is performed at mobile devices such as smartphones, which have constrained computing resource and power budgets. Consequently, the cost of modern neural channel decoder, if cannot be properly trimmed down, could severely impede the widespread adoption of this emerging solution.

**Technical Contributions.** In this work, we propose an efficient diffusion model-based decoding algorithm. Unlike existing diffusion decoder (Choukroun & Wolf, 2023), our method takes a different formulation of diffusion process via effectively integrating the philosophy of belief propagation into the model architecture, simultaneously achieving high error-correcting performance and low model complexity. More specifically, considering belief propagation is not noise oriented and its input format (log-likelihood ratio) needs a flexible noise design in the forward diffusion process, our proposed neural channel decoder is built upon the framework of variational diffusion model (Kingma et al., 2021), successfully reaping the benefits of belief propagation and diffusion process.

We summarize the contributions of this work as follow: 1) It, for the first time, studies the efficient integration of belief propagation-based channel decoding to diffusion model, formulating a new variational diffusion model based neural channel decoder. 2) The proposed variational diffusion decoder achieves ultra-low model complexity with high error-correcting performance. Compared to the state-of-the-art neural decoder, our approach brings $1000\times$ reduction in computational costs and memory costs, with the same or lower bit-error-rate (BER).

## 2    RELATED WORKS

To date a series of research efforts have been reported in applying neural networks to channel coding. In general, from the perspective of machine learning, by defining training loss as the binary cross entropy loss, the objective of channel decoding can be described as a multi-label binary classification problem, which has been well studied in deep learning community. Here different from many deep learning applications, the input messages of channel coding can be randomly generated, making the neural channel decoders free from data hungry concern (Marcus, 2018). It is also worth noticing that belief propagation-based deep learning models are found able to learn from all zero messages (Lugosch & Gross, 2017); while there has not been found any significant difference between training on all zero messages or random messages.

More specifically, neural channel decoders can be roughly categorized into belief propagation based and general neural network based. Belief propagation-based neural models maintain the message passing structure of BP algorithm, while adding neural weight parameters on messages (Nachmani et al., 2016; Lugosch & Gross, 2017; Liang et al., 2018; Nachmani & Wolf, 2019; Liao et al., 2021). Although these models provide performance improvement over classical BP approach, they are also constrained to the BP structure and are lack of the flexibility of neural architecture exploration. On the other hand, the structure of general neural network-based decoding models are designed without prior constraint. Gruber et al. (2017) propose the dense layer-based neural decoder, which works well for short channel codes with code length up to 64.

Instead of decoding the input message, Bennatan et al. (2018) take another path as directly predicting the transmission noise. To that end, it builds a binary-input symmetric-output channels-based framework to decouple message and noise. This strategy asks for additional information (syndrome) as input and needs extra computation for output, since the learning model is trained for predicting noise. In addition to these changes in processing input and output, the transition from message prediction to noise prediction is also relaxed to general neural network structure, enabling the flexible model design.

Based on the framework, Choukroun & Wolf (2022) successfully applies the transformer architecture in channel decoding, and this powerful architecture is later extended to the diffusion-based

model (Choukroun & Wolf, 2023). Considering denoising diffusion probabilistic models (DDPMs) Ho et al. (2020) naturally learn to predict noise, the neural decoder in Choukroun & Wolf (2023) is built in the format of DDPM. To accommodate the channel coding setting, the additive white Gaussian noise (AWGN) channel is formulated as an unscaled forward diffusion process. The reverse diffusion process is also modified into a decoding process, and the corresponding decoding steps (sampling timesteps in reverse process) are bounded by the parity check count in the given channel codes.

One key challenge for diffusion model-based channel decoder is the high complexity. Though powerful, diffusion models are known for their large computation cost and slow generation process, in addition to the underlying heavy deep learning model. For instance, the reverse timesteps in image generation can be as large as 1000 (Ho et al., 2020). To address this challenging issue, many research works have been proposed to improve the generation speed, such as reducing sampling steps (Song et al., 2021), transforming into using faster solvers for ordinary different equation (Lu et al., 2022) and knowledge distillation into a deterministic model (Salimans & Ho, 2022). These existing diffusion model optimization methods are orthogonal to our proposed solution of efficient variational diffusion channel decoder, and can be potentially applied to our approach towards further improvement.

# 3 BACKGROUND

## 3.1 CHANNEL CODING

For an $(N, K)$ channel code with code length as $N$ and information length as $K$, it specific code format is determined by a generator matrix $\mathbf{G} \in \{0, 1\}^{K \times N}$ and a parity check matrix $\mathbf{H} \in \{0, 1\}^{(N-K) \times N}$. Given $K$-bit input message $\mathbf{m}^b \in \{0, 1\}^K$, the encoded $N$-bit codeword $\mathbf{x}^b \in \{0, 1\}^N$ can be calculated via $\mathbf{x}^b = \mathbf{m}^b \mathbf{G}$ with all computations in binary domain. In general, $\mathbf{x}^b$ is transmitted over a noisy channel. The goal of channel coding is to recover the input message from the corrupted transmitted codeword at the receiver end. In practice, a systematic encoding approach is often adopted such that receiver can easily recover the message by taking the first $K$ bits from decoded results (Lin & Costello, 2004).

In the memoryless AWGN channel, the transmitted output $\mathbf{y}_s \in \mathbb{R}^N$ is simulated by $\mathbf{y}_s = \mathbf{x} + w_s \xi$ with $\xi \sim \mathcal{N}(0, \mathbf{I})$. Here, $\mathbf{x} \in \{-1, 1\}^N$ is the bipolar representation computed from $\mathbf{x} = 1 - 2\mathbf{x}^b$. The $w_s$ is determined by code rate $r = K/N$ and the channel signal-to-noise ratio (CSNR) in $s$-dB, i.e., $w_s = 1/\sqrt{2\frac{K}{N}10^{s/10}}$. While there are many different decoding algorithms, belief propagation (BP) has achieved tremendous success in channel codes. It treats the parity check matrix as a factor graph, and belief messages are iteratively propagated and updated over the graph.

More specifically, for a given parity check matrix, there are $N$ variable nodes and $N - K$ check nodes in the associated factor graph. First, define a compact representation of transmitted messages using log-likelihood ratio (LLR):

$$l_v = \text{LLR}(\mathbf{y}_s^v) = \log \frac{p(\mathbf{y}_s^v | \mathbf{x}_v = 1)}{p(\mathbf{y}_s^v | \mathbf{x}_v = -1)} = \frac{2}{w_s^2} \mathbf{x}_v + \frac{2}{w_s} \xi_v, \tag{1}$$

where $\mathbf{y}_s^v$ is the $v$-th value of $\mathbf{y}_s$. LLR-based belief propagation estimates the $v$-th bit of $\mathbf{x}$ by computing:

$$
\begin{aligned}
u_{c \to v} &= 2\left[ \prod_{v' \in M(c) \backslash v} \tanh\left(\frac{u_{v' \to c}}{2}\right) \right], \\
u_{v \to c} &= l_v + \sum_{c' \in N(v) \backslash c} u_{c' \to v}, \\
s_v &= l_v + \sum_{c' \in N(v)} u_{c' \to v},
\end{aligned}
\tag{2}
$$

where $M(\cdot)$ and $N(\cdot)$ denote the neighboring variables and check nodes, respectively. $\mathbf{x}_v$ can be determined by the sign of $s_v$. The expensive computation cost of hyperbolic tangent function in

$u_{c \to v}$ can be simplified into more implementation-friendly operations (Hu et al., 2001):

$$u_{c \to v} = \min_{v' \in M(c) \backslash v} |u_{v' \to c}| \prod_{v' \in M(c) \backslash v} \text{sign}(u_{v' \to c}), \tag{3}$$

where the $\text{sign}(\cdot)$ function returns the sign of input.

## 3.2 DIFFUSION MODELS

Sohl-Dickstein et al. (2015) proposes a diffusion based framework to model complex data distributions from a thermodynamics perspective. The essential idea is to gradually destroy input distribution in the forward diffusion process and learn a reverse process to model the distribution. Ho et al. (2020) shows DDPMs can effectively generate high quality images by a special parameterization method. Let $\mathbf{x}_0$ be the input data in distribution $q(\mathbf{x}_0)$. The forward diffusion process is a Markov chain with Gaussian noise added at each timestep:

$$q(\mathbf{x}_T, \ldots, \mathbf{x}_1) = \prod_{t=1}^{T} q(\mathbf{x}_t | \mathbf{x}_{t-1}), \quad \mathbf{x}_t = \sqrt{1 - \beta_t} \mathbf{x}_{t-1} + \sqrt{\beta_t} \xi_t, \quad \xi_t \sim \mathcal{N}(0, \mathbf{I}), \tag{4}$$

where $\beta_t$ for all $t$ are predefined hyper-parameters. With $\alpha_t = 1 - \beta_t$ and $\bar{\alpha}_t = \prod_{s=1}^{t} \alpha_t$, we can have $\mathbf{x}_t = \sqrt{\bar{\alpha}_t} \mathbf{x}_0 + \sqrt{(1 - \alpha_t)} \xi$. The reverse process learns the Gaussian distribution $p_\theta(\mathbf{x}_{t-1} | \mathbf{x}_t) = \mathcal{N}(\mu_\theta(\mathbf{x}_t, t), \sigma_t^2 \mathbf{I})$ to form the Markov chain:

$$p(\mathbf{x}_T, \ldots, \mathbf{x}_0) = p(\mathbf{x}_T) \prod_{t=1}^{T} p(\mathbf{x}_{t-1} | \mathbf{x}_t), \quad \mathbf{x}_{t-1} = \mu_\theta(\mathbf{x}_t, t) + \sigma_t \xi, \tag{5}$$

where $\theta$ describe model parameters and $\sigma_t$ is a function of $\beta_t$ for all $t$. The learning objective is simplified from the evidence lower bound (ELBO) to the KL divergence between $q(\mathbf{x}_{t-1} | \mathbf{x}_0, \mathbf{x}_t)$ and $p_\theta(\mathbf{x}_{t-1} | \mathbf{x}_t)$, which has an analytical form due to their Gaussian nature:

$$\theta^* = \arg\min_\theta \sum_{t>1} \mathbb{E}[\frac{1}{2\sigma_t^2} || \frac{\sqrt{\bar{\alpha}_{t-1}} \beta_t}{1 - \bar{\alpha}_t} \mathbf{x}_0 + \frac{\sqrt{\alpha_t}(1 - \bar{\alpha}_{t-1})}{1 - \bar{\alpha}_t} \mathbf{x}_t - \mu_\theta(\mathbf{x}_t, t) ||^2], \tag{6}$$

where efficient training is proposed to optimize at random timestep with stochastic gradient descent.

Given $\mathbf{x}_t$ can be expressed in $\mathbf{x}_0$ as in the forward process, the learning objective becomes a function of $\mathbf{x}_0$ and $\xi_t$. Thus, $\mu_\theta(\mathbf{x}_t, t)$ can be a model predicting either $\mathbf{x}_0$ or noise $\xi_t$, leaving the other term to be derived together with $\mathbf{x}_t$. As an example, DDPMs are designed to learn to predict noise, and $\mathbf{x}_0$ can be computed given the predicted noise and $\mathbf{x}_t$. In this way, learning objective can be minimized through the training process.

## 4 VARIATIONAL DIFFUSION CHANNEL DECODER

It can be noticed that the parameterization in DDPMs is specially designed, such as the relationship between $\alpha$ and $\beta$ and between $\mathbf{x_t}$ and $\mathbf{x}_0$. Although this specific design results in clear formulations for training and generation, such constraints are not realistic in channel coding, especially when describing the AWGN channel from the perspective of forward diffusion process. In this section, we propose an efficient decoding method using a flexible variational diffusion models (VDMs) framework (Kingma et al., 2021), i.e., variational diffusion channel decoder (VCDC).

### 4.1 AWGN AND FORWARD PROCESS

Different from DDPMs, VDMs generalize the mean and variance setting in the forward diffusion process. It enables the flexible Gaussian transition $q(\mathbf{x}_t | \mathbf{x}_0) = \mathcal{N}(\alpha_t \mathbf{x}_0, \sigma_t^2 \mathbf{I})$, without constraints on the relation between $\alpha_t$ and $\sigma_t$. We find such flexibility better help describe the AWGN channel as the forward diffusion process than DDPMs, especially with inputs in LLR format. For different AWGN channels in $T$ different CSNRs, we define the forward diffusion process by transmitting bipolar codeword $\mathbf{x}$ across these channels and the reverse process by denoising the transmitted messages at $t$-th CSNR to recover $\mathbf{x}$.

More specifically, let $\mathbf{z}_s = \text{LLR}(\mathbf{y}_s)$, and we have the distribution $q(\mathbf{z}_s|\mathbf{x}) = \mathcal{N}(\frac{2}{w_s^2}\mathbf{x}, \frac{4}{w_s^2}\mathbf{I})$, where $\alpha_s = \frac{2}{w_s^2}$ and $\sigma_s = \frac{2}{w_s}$. Let $\mathbf{z}_t$ be another channel in $t$-dB CSNR. According to VDM, the forward transition probability from $s$-dB channel to $t$-dB channel is:

$$q(\mathbf{z}_t|\mathbf{z}_s) = \mathcal{N}(\alpha_{t|s}\mathbf{z}_s, \sigma_{t|s}^2\mathbf{I}) = \mathcal{N}(\frac{\alpha_t}{\alpha_s}\mathbf{z}_s, (\sigma_t^2 - \alpha_{t|s}^2\sigma_s^2)\mathbf{I}), \tag{7}$$

where $\sigma_{t|s}^2$ should be positive since $s$ and $t$ are different:

$$\sigma_{t|s}^2 = \sigma_t^2 - \alpha_{t|s}^2\sigma_s^2 = (\frac{2}{w_t})^2 - (\frac{w_s^2}{w_t^2})^2(\frac{2}{w_s})^2 > 0 \implies w_t > w_s \implies t < s. \tag{8}$$

It is worth noticing that $s$ describes channel SNR rather than timestep. In this scenario, $s$-dB channel is at earlier timestep than $t$-dB channel, but it is found $t < s$. Given $T$ different channels in CSNRs $\{s_1, \ldots, s_T\}$, this observation requires $s_1 > \cdots > s_T$. In summary, the decreasing CSNR order is required in forward diffusion process.

Correspondingly, VDM defines SNR at $i$-th timestep by $\text{VSNR}(i) = \alpha_i^2/\sigma_i^2$ which we name as VSNR in this paper. VDM requires forward diffusion process with VSNR in a decreasing order. As $q(\mathbf{z}_{s_i}|\mathbf{x}) = \mathcal{N}(\frac{2}{w_{s_i}^2}\mathbf{x}, \frac{4}{w_{s_i}^2}\mathbf{I})$, $\text{VSNR}(i) = 1/w_{s_i}^2$ and $w_{s_i}$ is inversely proportional to $s_i$, the decreasing VSNR order is equivalent to the decreasing CSNR order, which aligns with our observation.

### 4.2 BELIEF PROPAGATION IN REVERSE PROCESS

While the noise based framework (Bennatan et al., 2018) is more flexible to deep learning models, it asks for more information (i.e., syndrome) as input and has a more complicated decoding process. Instead, traditional channel decoding methods like belief propagation are more straightforward, and they can predict $\mathbf{x}$ from $\mathbf{z}_t$ using only LLR. We propose to empower the belief propagation method with neural network to improve decoding performance; while keeping its low complexity.

Existing neural belief propagation (NBP) decoding algorithms (Nachmani et al., 2016) learn weight parameters on all message inputs in Eq. 2. As found in (Liao et al., 2021), such design can be over-parameterization given the similarity between $s_v$ and $u_{v \to c}$. It can be noticed that the difference between $s_v$ and $u_{v \to c}$ is simply $u_{c \to v}$. Therefore, learning neural parameters on message $u_{c \to v}$ can be sufficient for decoding:

$$s_v = u_{v \to c} + u_{c \to v}(\theta, u_{v \to c}) \implies \mathbf{x} \leftarrow \mathbf{x} + f(\mathbf{w}, \mathbf{x}), \tag{9}$$

where on the right side, the design is reformulated in the style of neural network layer formulation such that it becomes easy to understand and implement in current deep learning framework. $\mathbf{w}$ is the neural layer weight parameter, and $\mathbf{x}$ is the layer input. The first layer input is $l_v$ for all $v$. It should be noted that $f(\cdot)$ maintains the original $u_{c \to v}$ message formulation. To achieve high efficiency, each layer learns a shared weight parameter on all of its inputs.

We construct our deep learning model with $N - K$ layers as a neural block following Eq. 9, and the model complexity can be increased by stacking multiple blocks. In terms of computation cost, the reverse process can be treated as stacking shared weights models. Thus, adding new blocks increases complexity at intra-scale level while adding more reverse timesteps increases at inter-scale level. Given adding more timesteps maintains the same storage cost, it is more desirable to develop models by adding more timesteps for practical deployment purpose.

For the reverse diffusion process, we use $\hat{\mathbf{x}}_\theta(\mathbf{z}_t)$ to represent the model prediction given input $\mathbf{z}_t$ at timestep $t$. The related transition probability can be expressed as:

$$p(\mathbf{z}_s|\mathbf{z}_t) = q(\mathbf{z}_s|\mathbf{z}_t, \mathbf{x} = \hat{\mathbf{x}}_\theta(\mathbf{z}_t)) = \mathcal{N}(\mu_Q, \sigma_Q)$$
$$= \mathcal{N}(\frac{\alpha_{t|s}\sigma_s^2}{\sigma_t^2}\mathbf{z}_t + \frac{\alpha_s\sigma_{t|s}^2}{\sigma_t^2}\hat{\mathbf{x}}_\theta(\mathbf{z}_t; t), \frac{\sigma_{t|s}^2\sigma_s^2}{\sigma_t^2}\mathbf{I}) \tag{10}$$
$$= \mathcal{N}(\mathbf{z}_t + (\frac{2}{w_s^2} - \frac{2}{w_t^2})\hat{\mathbf{x}}_\theta(\mathbf{z}_t; t), [(\frac{2}{w_s})^2 - (\frac{2}{w_t})^2]\mathbf{I}).$$

In addition, the purpose of channel coding is more focused on denoising than generation. Choukroun & Wolf (2023) choose to skip the noise addition step in the reverse process which is often found in

Table 1: Negative natural logarithm BER results of reverse process at different timesteps. **Higher values mean better decoding performance**. Results are reported for CSNR 4-dB, 5-dB and 6-dB. The timestep of reverse process is mentioned in model name. For example, "Ours-20" means ours decoding results at reverse timestep 20.

| Code | (N, K) | Ours-1 | | | Ours-5 | | | Ours-10 | | | Ours-20 | | |
|------|--------|------|------|------|------|------|------|------|------|------|------|------|------|
| | | 4 | 5 | 6 | 4 | 5 | 6 | 4 | 5 | 6 | 4 | 5 | 6 |
| LDPC | (121, 60) | 3.98 | 5.82 | 8.75 | 4.71 | 7.48 | 11.89 | 5.1 | 8.14 | 12.98 | 5.24 | 8.55 | 13.21 |
| LDPC | (121, 70) | 4.72 | 6.95 | 9.94 | 6.01 | 9.48 | 14.41 | 6.33 | 10.04 | 15.42 | 6.59 | 10.36 | 15.42 |
| LDPC | (121, 80) | 5.24 | 7.64 | 10.65 | 6.8 | 10.49 | 15.46 | 7.27 | 11.15 | 16.79 | 7.48 | 11.65 | 17.2 |
| LDPC | (49, 24) | 4.54 | 6.06 | 8.24 | 5.48 | 7.43 | 10.31 | 5.86 | 7.83 | 11.07 | 5.8 | 7.96 | 11.2 |
| Polar | (128, 64) | 2.88 | 3.28 | 3.74 | 3.74 | 4.84 | 6.12 | 4.84 | 6.62 | 9.09 | 6.46 | 9.31 | 13.11 |
| Polar | (128, 86) | 3.41 | 3.92 | 4.55 | 4.48 | 5.75 | 7.4 | 5.45 | 7.1 | 9.25 | 6.56 | 8.87 | 11.85 |
| Polar | (128, 96) | 3.66 | 4.22 | 4.93 | 4.5 | 5.84 | 7.61 | 5.46 | 7.7 | 10.4 | 6.35 | 8.86 | 12.33 |
| Polar | (64, 32) | 2.86 | 3.28 | 3.76 | 4.4 | 5.44 | 6.68 | 5.36 | 6.58 | 8.54 | 6.31 | 8.61 | 10.96 |
| Polar | (64, 48) | 3.76 | 4.48 | 5.32 | 4.8 | 6.25 | 8.1 | 5.63 | 7.56 | 9.84 | 6.14 | 7.95 | 10.61 |
| CCSDS | (128, 64) | 4.43 | 6.11 | 8.37 | 6.23 | 9.82 | 14.39 | 6.98 | 10.95 | 16.54 | 7.61 | 11.68 | 17.01 |
| MACKAY | (96, 48) | 4.67 | 6.07 | 7.96 | 6.39 | 9.12 | 12.06 | 7.26 | 10.38 | 13.76 | 7.59 | 11.04 | 14.45 |

DDPMs. Besides, they also use number of parity checks in **H** as the maximum reverse timestep. We take a similar approach for reverse process except that we find their reverse timestep bound is loose because $N - K$ can often be unnecessarily large causing high complexity. In practice, we limit our reverse process up to 20 timesteps which already achieves competitive results. The number of parity check errors is also applied to perform early stopping during the reverse process.

## 5 EXPERIMENTS

We set CSNR from 4-dB to 6-dB as previous work (Choukroun & Wolf, 2023) adopts and evaluate our method on different linear block codes, i.e., Polar Codes (Arikan, 2009), Low-Density Parity Check (LDPC) codes (Gallager, 1962), Mackay codes and CCSDS codes, which are available on (Helmling et al., 2019). Modern channel coding has stringent latency and model size requirement. From the perspective of deployment, we evaluate and compare different models in experiments. Although deep learning-based decoders often benefit from the power of increasing model size, the resulting deployment cost is not feasible in the case of channel coding, e.g., millisecond-level latency tolerance (Rico & Merino, 2020). For the purpose of achieving low error rate, fast inference and small model size, we compare different models in their lightest setting over the bit error rate, computation cost in terms of floating-point operations (FLOPs), and storage cost in terms of bytes.

To achieve maximum efficiency, we build our VDCD models using a single block neural network and set $T = 20$ for reverse process. For model training, Adam optimizer (Kingma & Ba, 2015) is applied using learning rate 0.001 with 256 samples per batch and 20000 training iterations. Different from many diffusion models, the high efficiency of our model enables training and experiments on a CPU only platform, i.e., AMD EPYC 7402P 24-Core Processor. The training time varies between 1 to 3 hours for different codes.

Comparison is made with the hyper graph neural network-based model (Nachmani & Wolf, 2019) that also maintains the belief propagation structure, which is referred as HGN. Their fastest models are configured with hidden dimension 32 and 5 hidden layers. We also compare with the state-of-the-art DDECC (Choukroun & Wolf, 2023), where their fastest models are configured with 2 self attention layers in hidden dimension 32. The traditional belief propagation algorithm running 5 iterations is also listed as the baseline.

### 5.1 BIT ERROR RATE

We evaluate bit error rate (BER) for different channel codes and different models. For a given channel code, this metric shows the percentage of error bits in decoding results of different models. As bit error rates can be as low as $10^{-7}$, the evaluation only stops with at least 100 error samples detected. The test messages are randomly generated on the fly. Negative natural logarithm is taken to compare model performance because BER can differ in the order of magnitude. The higher result value means the lower BER, and it indicates better decoding performance.

Table 2: Negative natural logarithm BER results comparison between different models. **Higher values mean better decoding performance**.

| Code | (N, K) | BP | | | HGN | | | DDECC | | | Ours-20 | | |
|------|--------|-----|-----|------|-----|------|------|------|------|------|------|------|------|
| | | 4 | 5 | 6 | 4 | 5 | 6 | 4 | 5 | 6 | 4 | 5 | 6 |
| LDPC | (121, 60) | 4.82 | 7.21 | 10.87 | 5.22 | 8.29 | 13.0 | 4.48 | 6.95 | 10.65 | 5.24 | 8.55 | 13.21 |
| LDPC | (121, 70) | 5.88 | 8.76 | 13.04 | 6.39 | 9.81 | 14.04 | 5.41 | 8.22 | 12.22 | 6.59 | 10.36 | 15.42 |
| LDPC | (121, 80) | 6.66 | 9.82 | 13.98 | 6.95 | 10.68 | 15.8 | 6.12 | 9.38 | 13.25 | 7.48 | 11.65 | 17.2 |
| LDPC | (49, 24) | 5.3 | 7.28 | 9.88 | 5.76 | 7.9 | 11.17 | 5.27 | 7.38 | 10.23 | 5.8 | 7.96 | 11.2 |
| Polar | (128, 64) | 3.38 | 3.8 | 4.15 | 3.89 | 5.18 | 6.94 | 5.37 | 7.75 | 10.51 | 6.46 | 9.31 | 13.11 |
| Polar | (128, 86) | 3.8 | 4.19 | 4.62 | 4.57 | 6.18 | 8.27 | 5.61 | 7.76 | 10.42 | 6.56 | 8.87 | 11.85 |
| Polar | (128, 96) | 3.99 | 4.41 | 4.78 | 4.73 | 6.39 | 8.57 | 5.6 | 7.83 | 10.56 | 6.35 | 8.86 | 12.33 |
| Polar | (64, 32) | 3.52 | 4.04 | 4.48 | 4.25 | 5.49 | 7.02 | 5.99 | 8.16 | 10.9 | 6.31 | 8.61 | 10.96 |
| Polar | (64, 48) | 4.15 | 4.68 | 5.31 | 4.91 | 6.48 | 8.41 | 5.55 | 7.67 | 10.08 | 6.14 | 7.95 | 10.61 |
| CCSDS | (128, 64) | 6.55 | 9.65 | 13.78 | 6.99 | 10.57 | 15.27 | 5.79 | 8.48 | 12.24 | 7.61 | 11.68 | 17.01 |
| MACKAY | (96, 48) | 6.84 | 9.4 | 12.57 | 7.19 | 10.02 | 13.16 | 6.18 | 8.63 | 11.53 | 7.59 | 11.04 | 14.45 |

Table 3: FLOPs comparison between different models. DDECC models are diffusion models with reverse process. Column DDECC-1 lists FLOPs for DDECC models decoding for 1 reverse timestep. Column DDECC-Max lists FLOPs for DDECC models decoding with its maximum timesteps set as $N - K$ depending on given channel codes. Letters "K", "M" and "G" ending at each FLOPs number are data volume units standing for kilo, mega and giga, respectively.

| Code | (N,K) | BP | HGN | DDECC-1 | DDECC-Max | Ours-1 | Ours-20 |
|------|-------|-----|-----|---------|-----------|--------|---------|
| LDPC | (49, 24) | 54.1K | 34.1M | 269.67M | 6.7G | 3.5K | 70.4K |
| LDPC | (121, 60) | 316.4K | 1.6G | 2.3G | 140.3G | 18.9K | 377.6K |
| LDPC | (121, 70) | 263.8K | 920.4M | 2.1G | 107.1G | 15.7K | 314.8K |
| LDPC | (121, 80) | 211.1K | 476.1M | 1.91G | 78.3G | 12.6K | 251.6K |
| Polar | (64, 32) | 43.6K | 80.8M | 471.08M | 15.1G | 3.3K | 65.6K |
| Polar | (64, 48) | 45.0K | 30.4M | 370.28M | 5.9G | 3.0K | 60.4K |
| Polar | (128, 64) | 93.1K | 1.1G | 2.53G | 161.9G | 7.3K | 146.4K |
| Polar | (128, 86) | 141.9K | 935.0M | 2.11G | 88.6G | 9.6K | 192.4K |
| Polar | (128, 96) | 90.2K | 431.7M | 1.93G | 61.8G | 6.4K | 128.8K |
| CCSDS | (128, 64) | 161.9K | 562.0M | 2.53G | 161.9G | 10.2K | 204.8K |
| MACKAY | (96, 48) | 68.2K | 103.9M | 1.24G | 59.5G | 4.6K | 92.0K |

First, the BER change during the reverse process is studied. Table 1 lists BER values of our VCDC model for different codes at different timesteps. The first timestep is with the smallest BER, and BER can increase by adding more reverse steps. However, it can be seen that the BER increase becomes slower and tends to converge at last timestep. There are always positive BER increases with increasing timestep, which can come from either the continued diffusion process or the early stopping mechanism. Whenever the parity check error count goes to zero, the reverse process stops for the input sample. As a result, corresponding BER value of these samples remains the same for all timesteps. If the continued diffusion process improves BER, then overall BER results keep increasing. Therefore, positive increases between timesteps show the effectiveness of reverse process.

Table 2 shows the BER comparison between our model and other models. Our results come from the BER evaluation at reverse timestep 20 as shown in Table 1. It is found traditional belief propagation results can be strong baselines. In particular, our model results at timestep 1 from Table 1 are worse than belief propagation. DDECC model can also be worse than belief propagation, e.g., LDPC-(121, 60). However, HGN model results are consistently better than belief propagation. Overall, our model achieves the best decoding performance with the highest BER results compared with others. It can be also noticed that the BER change between SNRs are different among models. All deep learning model BER results consistently increase more for SNR $5 \to 6$ than SNR $4 \to 5$. But this does not hold for belief propagation with Polar-(128, 96) and Polar-(128, 64).

## 5.2 DECODER COMPLEXITY

We measure FLOPs and model size to represent the model computation cost and storage cost, respectively. Smaller numbers means more feasible for deployment. The model FLOPs is calculated

Table 4: Model size comparison between different models.

| Code | (N,K) | HGN | DDECC | Ours |
|---|---|---|---|---|
| LDPC | (49, 24) | 447.6KB | 131.6KB | 112.0B |
| LDPC | (121, 60) | 1.6MB | 226.3KB | 264.0B |
| LDPC | (121, 70) | 1.4MB | 218.2KB | 220.0B |
| LDPC | (121, 80) | 1.1MB | 210.0KB | 176.0B |
| Polar | (64, 32) | 596.3KB | 144.1KB | 316.0B |
| Polar | (64, 48) | 428.0KB | 135.9KB | 172.0B |
| Polar | (128, 64) | 1.4MB | 234.5KB | 752.0B |
| Polar | (128, 86) | 1.4MB | 217.6KB | 652.0B |
| Polar | (128, 96) | 1.0MB | 209.9KB | 444.0B |
| CCSDS | (128, 64) | 1.1MB | 234.5KB | 256.0B |
| MACKAY | (96, 48) | 647.7KB | 183.2KB | 192.0B |

with single input sample, i.e., batch size set as 1. The FLOPs of belief propagation is the FLOPs of total 5 message passing iterations. For diffusion models, FLOPs of single reverse timestep is measured, and the maximum FLOPs is computed by multiplying with the maximum timestep set for the model. DDECC sets the maximum reverse timesteps $N - K$, which can result in huge complexity. For our VCDC model, the maximum reverse timesteps is set as 20. Note that model size measurement is for models with weight parameters to store. Therefore, belief propagation model size results are not provided since they are all zeros.

Table 3 shows the number of FLOPs of different models. As expected, transformer based DDECC models have the largest FLOPs among all models because of the complicated model architecture. For a single timestep, DDECC models already take more computations than other models. Their computation cost increase linearly with the increase of reverse timesteps. HGN models make the second largest FLOPs with its graph neural networks. Our models are the fastest even compared with belief propagation because we apply the design of Eq. 3 and are smaller network models especially compared with others. This sacrifices the decoding performance at single timestep, since the BER results of our VCDC model at timestep 1 is worse than belief propagation. We are able to outperform other models by adding more timesteps, i.e., increasing our computation cost. When taking our model FLOPs of the whole reverse process (timesteps 20), our models cost 3 orders of magnitude less FLOPs than HGN models and 5 orders of magnitude than DDECC models. As a result, our models take slightly more FLOPs than belief propagation while achieving the best BER results.

Table 4 presents model sizes of different models. DDECC models are smaller than HGN models, but their FLOPS are more than HGN models. This shows DDECC models sacrifice more computation for less storage. Comparing the storage savings against computation savings, we argue it is harder to achieve storage savings. Unlike FLOPs amount, our models cost 3 orders of magnitude smaller storage than both HGN models and DDECC models. Our storage savings is the outcome of adding more reverse timesteps rather than adding more neural weight parameters. Even though adding more timesteps means longer reverse process, it hypothetically translates the reverse process into running one time inference over a deeper shared weight model.

## 6 CONCLUSIONS

This paper proposes a diffusion based decoder for channel coding. Driven by the strict requirements of low latency and high reliability, our model is designed to leverage traditional belief propagation and modern diffusion frameworks. The method reformulates the AWGN channel as forward diffusion process in the VDM framework and builds a neural network architecture on top of belief propagation. Experiments show our design achieves the best decoding performance with orders of magnitude savings in computation and storage cost than the state-of-the-art diffusion based decoder. These significant results exhibit the great potential of deploying diffusion based decoder in reality.

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
