# OpenReview forum: "VARIATIONAL DIFFUSION CHANNEL DECODING: A ULTRA-LOW-COST NEURAL CHANNEL DECODER"
_ICLR.cc/2025/Conference — Submitted to ICLR 2025_

### Official Review · Reviewer_t6tR · 2024-10-30

**Soundness:** 2
**Presentation:** 1
**Contribution:** 3
**Rating:** 5
**Confidence:** 4

**Summary:**

The paper presents an intriguing combination of recently proposed diffusion process-based channel decoding and traditional belief propagation methods to improve the efficiency of the decoding process. The proposed method parameterizes the LLR belief propagation steps using a neural network with shared parameters, incorporating a variational diffusion process for effective denoising. Experimental results demonstrate that the proposed method achieves superior decoding accuracy while maintaining good efficiency, almost comparable to that of the simple belief propagation approach.

**Strengths:**

1. The proposed method is well-motivated, and the experimental results support the claims.
2. The combination of diffusion-based models and belief propagation is interesting.
3. The proposed neural decoder architecture is highly efficient, even allowing for training on a CPU.

**Weaknesses:**

Although the proposed method is interesting, the paper is missing some important details. Each item may not be significant on its own, but together, they make it hard to grasp the necessary details, which makes me hesitant to advocate for the acceptance of the paper. I am happy to discuss and re-evaluate the soundness and presentation of the paper if the modifications are posted during the discussion period.

1. It is unclear why the VDM with $\alpha_s = \frac{2}{w_s^2}$ and $\sigma_s = \frac{2}{w_s}$ better describes the AWGN channel compared to the DDPM (a.k.a. variance preserving, $\alpha_s = \sqrt{1-\sigma_s}$) or the variance exploding ($\alpha_s = 1$) settings. Although most diffusion models use noise prediction, it is possible to let the model predict the original data $x_0$. Assuming the $x_0$ prediction setting, what are the major advantages of VDM over conventional diffusion processes?
2. More details regarding the implementation of the proposed method should be provided. What is the loss function used to train the proposed decoder? Regarding the structure, is the function $f$ in Eq. 9 composed of $N-K$ blocks? What type of activation is used in these blocks?
3. The results regarding bit error rate (BER) need better presentation. In Section 5.1, the negative log BER (the higher, the better) and the BER (the lower, the better) are used interchangeably, which creates confusion.
4. Additionally, in Tables 1 and 2, it is strange that the BER decreases as the message length $K$ increases while the code length $N$ is fixed. Intuitively, I would expect more errors at higher code rates. Please provide more information regarding this trend along $K$ in Tables 1 and 2.

**Questions:**

In addition to the questions raised in Weaknesses, I have the following questions:
1. What is the rationale behind using $N-K$ layers in a neural block?
2. I am curious if the proposed method scales up to decode long messages (e.g., $K > 64$).

---

### Official Review · Reviewer_S5F6 · 2024-10-31

**Soundness:** 2
**Presentation:** 1
**Contribution:** 2
**Rating:** 3
**Confidence:** 5

**Summary:**

The paper proposes to extend the original DDECC framework by adapting VDM with the neural BP algorithm.

The method is shown to benefit from the low complexity of BP as well as from the low capacity of neural BP and seems to provide very good decoding performance.

**Strengths:**

The idea is quite original: the idea to use neural BP, while natural, requires adaptation of the original DDPM via VDM.

The improvement seems substantial.

**Weaknesses:**

1 - Presentation :

The paper dramatically lacks clarity.
The method is not clearly described and no code is available. VDM should be transferred/added to the background.

The neural decoder used is not described.

There is plenty of room (2 pages) for improving the paper such that Algorithms and visual depictions of performance on more SNRs would be appreciated.

2 - Literature:

The authors refer to Bennatan et al. as a noise-predicting framework "enabling more relaxed model design" while its only purpose is to provide a new decoding procedure free from overfitting for model-free solutions.

Recent references (neural decoders) are missing [1,2,3,4].

3 - Novelty:
As far as I know, DDECC is presented as a decoding framework for model-free decoders and the choice for the ECCT is because of its SOTA performance. Thus the proposed method extends it to model-based decoders. More specifically, the DDECC does not impose scaling or scheduling constraints.

The use of VDM vs DDPM should be ablated (it's straightforward to get the noise from the prediction).



4- Results:

The choice of the baselines is not clear at all, and their performance with different capacities should be presented.
While BP is already a very well-optimized decoder, model-free decoders and larger neural BP (e.g., HGN) may benefit from heavy compressions such that choosing very shallow ("fastest") decoders is also unfair. For example, the ECCT is almost as sparse as BP [1] but the number of FLOPs in the masked Transformer won't take this into account. A complexity analysis is then better suited here.

More specifically, DDECCT converges within very few iterations (1-3) such that comparison with 20 BP iterations is not right.

Comparison with SCL on Polar codes is also natural. Comparison with SOTA neural BP [4] is missing.

Thus the claims of superiority should be moderated or much better established.

[1] CrossMPT: Cross-attention Message-Passing Transformer for Error Correcting Codes, Park et al.

[2] A Foundation Model for Error Correction Codes, Choukroun and Wolf

[3] Learning Linear Block Error Correction Codes, Choukroun and Wolf

[4] Autoregressive Belief Propagation for Decoding Block Codes, Nachmani and Wolf

**Questions:**

While I believe the adaptation/variation of the DDECC to model free decoders is important, the weaknesses stated above drastically deteriorate the paper's claims.

1 - Please address the Weaknesses above.

2- Minor typo:

line  244: such design can be *overparameterization* ?

---

### Official Review · Reviewer_96k5 · 2024-11-01

**Soundness:** 1
**Presentation:** 2
**Contribution:** 2
**Rating:** 1
**Confidence:** 4

**Summary:**

The paper proposes the use of a neural belief propagation algorithm and a reverse diffusion process for error-correcting code (ECC) decoding.

**Strengths:**

The studied problem is practically significant in communication systems

**Weaknesses:**

The related work section could be improved, other ways of solving the problem are not acknowledged.

The architecture choice (variational diffusion) could be better justified, there are many newer architectures specifically designed for low-complexity inference.

Latency measurements could be added, given that they are practically relevant in communication systems.

**Questions:**

The presentation of the method is lacking, and it is not clear if and how belief propagation is interleaved with the reverse diffusion process at all. The signals introduced in (9) are not present in equation (10).

The machine learning novelty is limited. The methods are empirical and applied to a specific problem (ECC decoding), which would generally be much more suitable for a domain-specialized conference or journal.

The space of ECC decoding using deep learning has been extensively studied in the past years and the related work is significantly lacking.

Entire classes of approaches are not acknowledged or compared against, such as unrolled/unfolded methods [1] or joint denoising (detection) and detection methods [2]. Some of these methods intentionally use very small neural networks (e.g., [2] uses only two layers) for complexity reasons.

There is also a vast literature body on accelerating diffusion sampling via distillation - see the blog post [3] for a survey of eight modern methods.

None of these techniques are compared against or discussed, which is a major drawback of the submission given that complexity reduction is the focus.

Why was variational diffusion specifically chosen instead of more recent architectures (where architecture = learning formulation and ODE parameterization) with much better practical results on few-step samplers - such as EDMs, consistency models, or rectified flow models?

There are also diffusion architectures that are specifically designed to give good results on few-step results on denoising and inverse problem, e.g. as in [4] and its many extensions.

Despite acknowledging latency as a key issue, there are no latency measurements presented. Given that commodity CPUs and GPUs are in the 200-450 W range and already over-powered for communications, it would have strengthened the submission if latency figures were measured for all methods.

It would be even more relevant to measured the required latency to reach a certain bit error rate and show how this scales.

From a practical communication system point of view, it seems all derivations and methodology only works for bipolar (BPSK) signals x. Practical systems may also use modulation (e.g., 16-QAM with Gray mapping), which would change the expressions of the conditional distribution for each bit transmitted in a single channel use. It is not fully clear if the current derivations are easily extendible to that case.

---

### Meta-Review · Area_Chair_szLp · 2024-12-20

**Metareview:**

This paper proposed a method for channel decoder using variational diffusion models. The key idea is to integrates the traditional belief propagation with diffusion models. However, all reviewers, including my own reading, agree that the paper is below the acceptance level with the following main concerns:

1.  Poor presentation.

2.  Limited novelty.

3.  Insufficient experiments.

4. Lack of some highly related references.

As a result, based on the above weaknesses raised by most reviewers, I would recommend rejection.

**Additional Comments On Reviewer Discussion:**

No rebuttal was submitted by the authors.

---

### Decision · Program_Chairs · 2025-01-22

Reject